# Fatigue, Depression, and Anxiety in Patients with COPD, Asthma and Asthma-COPD Overlap

**DOI:** 10.3390/jcm11247466

**Published:** 2022-12-16

**Authors:** Henryka Homętowska, Jakub Klekowski, Natalia Świątoniowska-Lonc, Beata Jankowska-Polańska, Mariusz Chabowski

**Affiliations:** 1Specialist Hospital of the Ministry of the Interior and Administration, 40 Karłowicza Street, 48-340 Głuchołazy, Poland; 2Student Research Group No 180, Faculty of Medicine, Wroclaw Medical University, 50-367 Wrocław, Poland; 3Division of Anesthesiological and Surgical Nursing, Department of Nursing and Obstetrics, Faculty of Health Science, Wroclaw Medical University, 5 Bartla Street, 51-618 Wroclaw, Poland; 4Innovation and Research Center, 4th Military Teaching Hospital, 5 Weigla Street, 50-981 Wrocław, Poland; 5Department of Surgery, 4th Military Teaching Hospital, 5 Weigla Street, 50-981 Wrocław, Poland

**Keywords:** respiratory diseases, COPD, asthma, asthma-COPD overlap, depression, fatigue

## Abstract

Introduction. Asthma and COPD are extremely common respiratory diseases that have a serious impact on people’s lives around the world. A disease characterized by symptoms characteristic for asthma and COPD is called asthma-COPD overlap (ACO). Fatigue and certain psychological disorders such as anxiety and depression are important comorbidities in these diseases. The purpose of this study was to assess the prevalence of fatigue, anxiety, and depression in patients with asthma, COPD, and ACO and to also consider their mutual correlations. Material and Methods. A total of 325 patients were enrolled in the study. There were 159 women and 166 men and their mean age was 63. Two standardized questionnaires were used: the Modified Fatigue Impact Scale (MFIS) and the Hospital Anxiety and Depression Scale (HADS). Results. The mean total MFIS score for all patients was 33.03. Patients with asthma generally scored lower than patients with COPD and ACO. There were no statistical differences in the HADS for anxiety between the groups, although around half of the patients registered a score indicating some level of disorder. Patients with COPD and ACO were proven to suffer more from depression than patients with asthma. The HADS and MFIS scores were found to correlate significantly and positively. Conclusions. Our study showed that patients with COPD, asthma, and ACO generally suffered from an increased level of fatigue and depression. Anxiety was high in all groups, but it was at a similar level for patients suffering from each of the three diseases under consideration. It is important to treat the physical symptoms as well as the psychological disorders since they greatly impact on the patient outcomes.

## 1. Introduction

Chronic obstructive pulmonary disease (COPD) and asthma are common worldwide diseases that are characterized by an inflammatory process of the airway, resulting in an obstruction and the limitation of airflow. Smoking is generally linked to COPD as a primary risk factor. Patients with COPD suffer from persistent symptoms such as a chronic cough, breathlessness, and excessive sputum production. COPD has become the fourth most common cause of death globally and is predicted to be the fifth most common cause of disability [1,2,3]. Asthma typically coexists with allergies and the symptoms are generally a result of airway hyperresponsiveness, resulting in bronchial contraction and obstruction of the airflow. Asthma usually presents as episodes or attacks. Symptoms such as wheezing, shortness of breath, coughing, and tightness of the chest are often triggered by different factors, for example, respiratory tract infections, low temperatures, cigarette smoke, air pollution, allergens, stress, exercise, and other factors. The etiology of asthma is not certain, but genetic factors are believed to play an important role [1,4,5].

Although COPD and asthma seem to have distinctive differences in their symptoms, which should allow specialists to easily differentiate these two diseases, this is not always the case. There are patients who have features of both asthma and COPD, and this situation is as designated to be asthma-COPD overlap (ACO). The Global Initiative for Asthma (GINA) and the Global Initiative for Chronic Obstructive Lung Disease (GOLD) collectively prepared a definition describing this phenomenon as a clinical condition characterized by the persistent limitation of airflow with coexisting features of asthma and COPD [4,6,7].

The respiratory symptoms of the above-mentioned conditions have been widely described and extensively assessed in patients, but there are emerging studies that point out important but often overlooked problems such as fatigue, anxiety, and depressive disorders, which greatly impact the patients’ lives. Fatigue is a notably subjective symptom described by patients as a general tiredness and lack of energy, which prevents patients from performing daily activities. Fatigue often occurs in the course of respiratory diseases as a result of labored breathing. Patients with asthma, COPD, or ACO might experience fatigue of various degrees and frequency depending on the primary disease. It is reported to be the main extrapulmonary symptom that affects up to 70–95% of COPD patients and worsens their prognosis. In asthma, fatigue is mostly associated with episodes of exacerbations, while patients with COPD tend to experience fatigue on a daily basis. Psychosocial factors are important aspects of COPD. Due to the chronic and progressing character of the disease, patients are not only physically limited, but also often experience diminished psychological and social functioning, which could implicitly affect the course of the disease and patient’s social environment. Anxiety is a common psychological dysfunction among patients with asthma, with a prevalence of 16–52%. Similarly, up to 40% of patients with COPD experience anxiety. Depressive disorders are no less important and studies report a 25% and even up to 41% occurrence of depression in COPD and asthma, respectively. It is important to take the comorbidity of psychological disorders into consideration with somatic diseases, since it is evident that they decrease the patients’ quality of life [2,3,8,9,10,11,12].

The purpose of this study was to assess the prevalence of fatigue, anxiety, and depression in patients with asthma, COPD, and ACO and to also consider their mutual correlations.

## 2. Methods

The study was designed as a prospective questionnaire-based study. It included 325 patients who had been diagnosed with asthma, COPD, or ACO. The inclusion criteria were: informed written consent to participate in the study, diagnosis of COPD or asthma or ACO, age above 18 years. Patients below 18 years, with concomitance of severe chronic diseases that could interfere with the study (neoplastic diseases of the lungs, respiratory failure, exacerbation of chronic disease), and unable to complete the questionnaire independently were excluded from the study. The clinical and sociodemographic data were collected from self-reported questionnaires and the patients’ medical records. All of the patients gave informed written consent to participate in the study and to answer the questionnaires. The study was approved by the Bioethics Committee at Wroclaw Medical University (KB 737/2018) and was conducted according to the standards of the Declaration of Helsinki.

The patients were asked about their age, sex, place of residence (country, city), marital status, employment status, tobacco use, number of complications, the duration of their disease, and the asthma self-control methods they employed. Two standardized questionnaires were used to assess the patients’ levels of fatigue, anxiety, and depression: the Modified Fatigue Impact Scale (MFIS) and the Hospital Anxiety and Depression Scale (HADS).

The MFIS consists of 21 items divided into three subscales: physical, cognitive, and psychosocial functioning. The scoring ranges from 0 to 84 points with higher values indicating greater fatigue. The scale was originally meant to assess the level of fatigue and its impact on patients with multiple sclerosis. Cronbach’s alpha for the MFIS is 0.92, which means the scale has good internal consistency. The MFIS has been validated for use in Poland [13,14,15,16].

The HADS was developed to evaluate symptoms of depression among patients receiving general medical care. The scale consists of seven items, each of which can be scored from 0 to 3 with a total score ranging from 0 to 21. Scores below 8 imply the absence of disorders, scores between 8 and 10 suggest a ‘borderline abnormal’, and scores above 10 indicate ‘abnormal’, probable depression/anxiety. The scale has been validated for use in Poland [17,18,19,20].

Statistical analysis. The analysis of quantitative variables was conducted by calculating the means, the median, quartiles, the minimum, and the maximum. The analysis of the qualitative variables was carried out by calculating the number and percentage of occurrences of each value. The comparison of the qualitative variables in groups was shown using the chi-square test or Fisher’s exact test. The comparison of quantitative variables in pairs of groups was carried out using the Mann–Whitney’ test. The comparison of quantitative variables in three or more groups was carried out with the Kruskall–Wallis test and post hoc analysis with the Dunn test. Correlations between quantitative variables were analyzed using Spearman’s correlation coefficient. A *p*-value below 0.05 was considered statistically significant. The statistical analysis was carried out with version 3.6.2 of the R program.

## 3. Results

The mean age of the 325 patients enrolled in the study was 63.04 (SD = 11.29) and ranged between 21 and 96. There were 159 women (48.92%) and 166 men (51.08%). A total of 142 (43.69%) of the patients had been diagnosed with COPD, 109 (33.54%) with asthma, and 74 (22.77%) with ACO. Most of the patients were in a relationship (73.54%), only 20.62% had higher education, and almost two-thirds (63.08%) lived in a city. Only slightly over 33% were professionally active. A total of 25.54% of the patients declared themselves to be regular smokers. The groups of patients with specific diseases differed statistically from each other in several domains. Patients with COPD were significantly older than the other patients. There were statistically more women among the asthmatics than in the other groups. A total of 70% of the COPD patients were retired, which was significantly more than in the other groups (40–50%). The largest number of regular smokers were patients with ACO, while the fewest smokers were in the group of asthmatics. The same pattern could be seen in the relationship to the number of hospital admissions due to complications. Patients with asthma claimed to be better able to self-control their disease than patients with ACO. The detailed sociodemographic characteristics of the group who participated in the study are presented in a Table 1.

The mean total MFIS score for all patients was 33.03 and the median was 33. The patients with ACO had significantly higher scores in total and on the physical function subscale. On the psycho-social subscale, patients with COPD and ACO scored higher than patients with asthma (Table 2).

Age was found to correlate significantly and positively with the total MFIS score and with the cognitive and psycho-social subscales. Cognitive functions were worse among the pensioners and retired than among those in employment. Regular smokers were found to have significantly higher scores on the physical and psycho-social subscales. The total MFIS score proved to be higher in patients suffering from two or more complications within a year. Patients who had been struggling with the disease for between 5 and 10 years turned out to have significantly higher scores on the psycho-social subscale than patients who had the disease for longer than 10 years. The self-control of asthma proved to have a positive impact on fatigue because patients claimed to have good control over their symptoms scored lower on the MFIS.

From the HADS score for anxiety 54% of patients were below 8 points, 25% between 8 and 10, and 20% scored 11 and higher. A total of 59% of asthmatics revealed no symptoms of anxiety, as did 50% of patients with COPD and ACO. Consequently, about 40–50% of patients experienced more or less intense symptoms of anxiety. Nevertheless, there were no significant differences between the groups.

On the depression scale, almost 65% of patients scored within normal values, 26% were classified as being borderline abnormal, and 9% had scores indicating depression. In general, the patients with COPD and ACO scored statistically higher for depression. As many as 50% of patients suffering from COPD were classified as being ‘borderline abnormal’ or have serious symptoms of depression. This percentage is notably lower in patients with ACO (36.5%) and asthma (23.24%). These differences were statistically significant (Table 3 and Table 4).

The HADS for anxiety and depression were found to correlate significantly and positively with the total MFIS score and with the scores on each of its subscales, which indicates that the higher the HADS score, the greater the value in the MFIS (Table 5 and Figure 1).

Men scored significantly higher for depression (*p* < 0.001). Similarly to the MFIS, age correlated significantly and positively with the HADS score (anxiety r = 0.175, *p* = 0.002; depression r = 0.237, *p* < 0.001). Anxiety was found to be higher among pensioners and retired people (*p* = 0.009), while pensioners proved to be more depressed than employed or retired employees (*p* = 0.002). Furthermore, retired people scored significantly higher for depression than employed people. Significant correlations were found for smoking; depression was higher among regular smokers than non-smokers (*p* < 0.001), but anxiety was proven to be significantly higher for occasional smokers than non-smokers (*p* = 0.042). Patients who had experienced two and more disease complications regarding their condition in the last year were significantly more depressed (*p* = 0.004). A disease duration of over 10 years was associated with higher anxiety (*p* = 0.004) and depression (*p* = 0.041) than that found among patients who had the disease for up to 5 years.

## 4. Discussion

Our study has shown that patients with COPD, asthma, and ACO generally suffer from an increased level of fatigue. Patients with ACO reported higher levels of fatigue, especially on the physical subscale, which highlights the importance of proper symptom and disease control in those patients to ensure that they have a better capacity for daily activities. Approximately 45% of those questioned tended to have moderate and noticeable symptoms of anxiety, while about 35% suffered from depression to various degrees, which proves that these disorders are highly prevalent within the diseases under consideration. Taking into account all of the above suggestions that overall patients with COPD and ACO are more prone to suffer from disorders not directly linked to their disease such as anxiety and depression, it should also be remembered that their lives are greatly disrupted by a high level of fatigue. This implies a multidimensional approach to the management of these diseases in order to provide a better quality of life. The correlation between the HADS and MFIS scores only serves to strengthen the previous statement, as coping with fatigue, anxiety, and depression might simultaneously successfully relieve these disorders since they reinforce one another. Although in the face of previous findings the importance of thorough care in asthma may seem to be diminished, it is obvious that it should not be neglected, but a higher level of intervention appears to be of the essence in COPD and ACO.

Two studies recently conducted in China prove that anxiety and depression are extremely common comorbidities among patients with COPD. Both studies used the HADS score to assess the disorders. Zhao et al. questioned 72 patients with COPD and they reported that 38% had depression and 23% had anxiety. Additionally, the authors found that anxiety and depression were independent mutual risk factors and that they correlated positively with each other. Huang et al. studied 4686 patients with COPD, of whom 10.79% suffered from anxiety, 13.65% had depression, and 7.08% experienced anxiety and depression simultaneously. These studies are a reminder that it is crucial to detect anxiety and depression early, since these disorders are associated with worse outcomes [21,22]. Husain et al. similarly assessed anxiety and depression in 293 COPD patients in Pakistan, although they utilized alternative questionnaires: the Patient Health Questionnaire-9 (PHQ-9) and the Generalized Anxiety Disorder-7 (GAD-7). The prevalence of anxiety and depression in this study was 20% and 51%, respectively. Some further conclusions emerging from this study underline that depressed patients with COPD experience low social support, a worse quality of life, and greater social stress [23]. Fatigue (measured with the Checklist Individual Strength—CIS-Fatigue) was proven to be a major issue in COPD. It was estimated that 26% of patients experienced mild fatigue, while 48.5% suffered from severe fatigue. Women were found to have higher mean fatigue scores as well as a higher proportion of severe fatigue than men. Some factors such as smoking, medical complications, dyspnea, and worse lung function were associated with a higher risk of fatigue [24]. There are numerous studies exploring possible interventions targeting improvements in levels of anxiety and depression. Pulmonary rehabilitation, controlled breathing techniques, mind–body exercises, and physical activity are just some examples of the measures that can be taken in order to help reduce anxiety and symptoms of depression [25,26,27,28,29].

Ciprandi et al. questioned outpatients with asthma using the HADS and found that 36.9% of them were anxious and 11% were depressed, which was slightly lower than in our study. However, the mean HADS scores were 6.4 for anxiety and 3.8 for depression, which were close to the results of our analysis. Poor asthma control was a negative predictor of depression among patients in the cited study [30]. Another study explored the same topic, but with different questionnaires: the Beck Depression Inventory-II (BDI-II) and the Beck Anxiety Inventory (BAI). According to this study, anxiety occurred in 54.3% of patients, while the prevalence of depression reached 50.6% [31]. A study by Robinson et al. agreed that patients scoring low in asthma specific questionnaires such as the Asthma Control Questionnaire (ACQ) and the Mini-Asthma Quality of Life Questionnaire (mini-AQLQ) should be further screened for depression and anxiety [32].

Studying ACO and non-ACO cohorts, Yeh et al. proved that the risk of incident depression was higher in patients diagnosed with ACO, even in those with no comorbidities and in young patients. Anxiety was estimated to affect 10.5% of ACO patients [33]. The rate of anxiety among patients with the ACO in another study was 7.1% [34]. Van Boven et al. reported that 37.7% and 2.6% of ACO patients suffered from anxiety and depression, respectively [35]. These rates differ greatly from the findings of our study, but the reported data were from cohort studies and the methods used to assess them were not certain, so they may vary.

Limitations of the study. Our study provided us with an insight into depression, anxiety, and fatigue among patients with asthma, COPD, and ACO, but despite the strengths and valuable results, it had some weaknesses that could be reshaped into targets for further studies. This study does not present the correlations between the HADS and MFIS for each of the diseases, which we believe might be the main disadvantage of our work. Determining the percentages of patients suffering from anxiety and depression concomitantly would also make it possible to draw more precise conclusions. This work does not specify the severity of the diseases by spirometry, which could allow us to classify the patients more accurately. We expect that these unsolved parts of our study will soon be explored in forthcoming research.

## 5. Conclusions

The results suggest a high susceptibility to anxiety and depression among patients with ACO and COPD. Although patients with asthma are less afflicted by fatigue, anxiety, and depression, their needs must not be neglected nor their symptoms overlooked. Multidimensional approaches should be employed to deal with the above-mentioned problems, since this study confirms the positive correlation between HADS and MFIS.

## Figures and Tables

**Figure 1 jcm-11-07466-f001:**
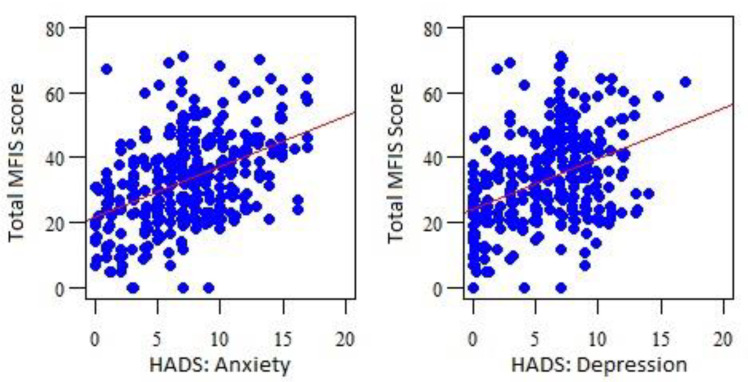
MFIS and HADS correlations. MFIS—Modified Fatigue Impact Scale. HADS—Hospital Anxiety and Depression Scale.

**Table 1 jcm-11-07466-t001:** Sociodemographic characteristics.

		Disease				*p*
		Asthma—A	COPD—B	ACO—C	Total	
Age [years]	Mean ± SD	58.8 ± 11.92	69.15 ± 8.7	62.2 ± 9.4	63.04 ± 11.29	*p* < 0.001 *
	Median	61	69	62	64	
	Quartiles	53–66	64.75–76	54–68	57–70	B > C, A
Sex	Women	106 (74.65%)	28 (25.69%)	25 (33.78%)	159 (48.92%)	*p* < 0.001 **
	Men	36 (25.35%)	81 (74.31%)	49 (66.22%)	166 (51.08%)	
Marital status	No relationship	45 (31.69%)	24 (22.02%)	17 (22.97%)	86 (26.46%)	*p* = 0.168 **
	In a relationship	97 (68.31%)	85 (77.98%)	57 (77.03%)	239 (73.54%)	
Education	Higher	42 (29.58%)	16 (14.68%)	9 (12.16%)	67 (20.62%)	*p* < 0.001 **
	Secondary	69 (48.59%)	43 (39.45%)	29 (39.19%)	141 (43.38%)	
	Vocational school	23 (16.20%)	37 (33.94%)	27 (36.49%)	87 (26.77%)	
	Primary	8 (5.63%)	13 (11.93%)	9 (12.16%)	30 (9.23%)	
Place of residence	City	96 (67.61%)	65 (59.63%)	44 (59.46%)	205 (63.08%)	*p* = 0.329 **
	Country	46 (32.39%)	44 (40.37%)	30 (40.54%)	120 (36.92%)	
Professional status	Active	48 (33.80%)	9 (8.26%)	26 (35.14%)	83 (25.54%)	*p* < 0.001 ***
	Active retired	11 (7.75%)	11 (10.09%)	3 (4.05%)	25 (7.69%)	
	Retired	60 (42.25%)	68 (62.39%)	29 (39.19%)	157 (48.31%)	
	Pensioner	15 (10.56%)	18 (16.51%)	15 (20.27%)	48 (14.77%)	
	Student	1 (0.70%)	0 (0.00%)	0 (0.00%)	1 (0.31%)	
	Unemployed	7 (4.93%)	3 (2.75%)	1 (1.35%)	11 (3.38%)	
Smoking	Regular	13 (9.15%)	32 (29.36%)	38 (51.35%)	83 (25.54%)	*p* < 0.001 **
	Occasionally	10 (7.04%)	30 (27.52%)	7 (9.46%)	47 (14.46%)	
	Non-smoker	119 (83.80%)	47 (43.12%)	29 (39.19%)	195 (60.00%)	
Number of hospital stays due to disease complications	Not once	47 (33.10%)	14 (12.84%)	10 (13.51%)	71 (21.85%)	*p* < 0.001 ***
	Once	52 (36.62%)	46 (42.20%)	27 (36.49%)	125 (38.46%)	
	2–3 times	38 (26.76%)	45 (41.28%)	30 (40.54%)	113 (34.77%)	
	4–5 times	3 (2.11%)	4 (3.67%)	7 (9.46%)	14 (4.31%)	
	More than 5 times	2 (1.41%)	0 (0.00%)	0 (0.00%)	2 (0.62%)	
Disease duration	Up to one year	7 (4.93%)	1 (0.92%)	1 (1.35%)	9 (2.77%)	*p* = 0.333 ***
	1–4 years	33 (23.24%)	20 (18.35%)	15 (20.27%)	68 (20.92%)	
	5–10 years	49 (34.51%)	47 (43.12%)	26 (35.14%)	122 (37.54%)	
	Over 10 years	53 (37.32%)	41 (37.61%)	32 (43.24%)	126 (38.77%)	

* *p*—Kruskal–Wallis test + post hoc analysis (Dunn’s test); ** *p*—chi-square test; *** *p*—Fisher’s exact test, statistically significant *p* < 0.05.

**Table 2 jcm-11-07466-t002:** MFIS scores: mean + SD, median, quartiles.

MFIS		Disease			*p*
		Asthma—A (N = 142)	COPD—B (N = 109)	ACO—C (N = 74)	
Total MFIS	Mean ± SD	30.19 ± 14.33	33.28 ± 13.53	38.14 ± 12.77	*p* < 0.001 *
	Median	29.5	34	37	
	Quartiles	21–39	23–43	29–46	C > B, A
Physical functions	Mean ± SD	17.55 ± 7.22	19.29 ± 7.53	23.85 ± 6.7	*p* < 0.001 *
	Median	18	19	24	
	Quartiles	13–23	14–24	20–29	C > B, A
Cognitive functions	Mean ± SD	10.13 ± 7.33	11.02 ± 7.2	10.81 ± 8.04	*p* = 0.552
	Median	10	10	10	
	Quartiles	4–15	5–16	5–17	
Psychosocial functions	Mean ± SD	2.51 ± 1.9	2.96 ± 1.75	3.47 ± 1.96	*p* = 0.002 *
	Median	2	3	3.5	
	Quartiles	1–4	2–4	2–5	C, B > A

* *p*—Kruskal–Wallis test + post hoc analysis (Dunn’s test), statistically significant *p* < 0.05.

**Table 3 jcm-11-07466-t003:** HADS scores: mean + SD, median, quartiles.

HADS		Disease			*p*
		Asthma—A (N = 142)	COPD—B (N = 109)	ACO—C (N = 74)	
Anxiety	Mean ± SD	6.85 ± 3.96	7.47 ± 4.04	7.8 ± 4.03	*p* = 0.167
	Median	7	7	7.5	
	Quartiles	4–9.75	4–10	6–10	
Depression	Mean ± SD	4.37 ± 3.52	6.89 ± 3.88	6.38 ± 3.38	*p* < 0.001 *
	Median	4	8	7	
	Quartiles	1–7	5–10	4.25–9	B, C > A

* *p*—Kruskal–Wallis test + post hoc analysis (Dunn’s test), statistically significant *p* < 0.05.

**Table 4 jcm-11-07466-t004:** HADS anxiety and depression—score proportion.

HADS		Disease			*p*
		Asthma (N = 142)	COPD (N = 109)	ACO (N = 74)	
Anxiety	Absence of disorders	84 (59.15%)	55 (50.46%)	37 (50.00%)	*p* = 0.425
	Borderline abnormal	30 (21.13%)	33 (30.28%)	19 (25.68%)	
	Abnormal	28 (19.72%)	21 (19.27%)	18 (24.32%)	
Depression	Absence of disorders	109 (76.76%)	54 (49.54%)	47 (63.51%)	*p* < 0.001 *
	Borderline abnormal	28 (19.72%)	35 (32.11%)	22 (29.73%)	
	Abnormal	5 (3.52%)	20 (18.35%)	5 (6.76%)	

* *p*—chi-square test or Fisher’s exact test, statistically significant *p* < 0.05.

**Table 5 jcm-11-07466-t005:** Correlations between the MFIS and HADS scores.

MFIS	HADS: Anxiety	HADS: Depression
	Spearman’s Correlation Coefficient	
Total MFIS	r = 0.43, *p* < 0.001 *	r = 0.395, *p* < 0.001 *
Physical functions	r = 0.235, *p* < 0.001 *	r = 0.166, *p* = 0.003 *
Cognitive functions	r = 0.485, *p* < 0.001 *	r = 0.497, *p* < 0.001 *
Psychosocial functions	r = 0.324, *p* < 0.001 *	r = 0.351, *p* < 0.001 *

* statistically significant *p* < 0.05. MFIS—Modified Fatigue Impact Scale. HADS—Hospital Anxiety and Depression Scale.

## Data Availability

All the data are available from the corresponding author upon request.

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
