# Peer review of "Fatigue, Depression, and Anxiety in Patients with COPD, Asthma and Asthma-COPD Overlap"

_jcm, 2022, doi:10.3390/jcm11247466_

Round 1

Reviewer 1 Report

The results presented a needs to be clearer, please see the highlighted text and comments it in the document.  Please make the definitions and the symptoms clear, also make sure the words used are appropriate in the text For example, page 1 line 44: 'Asthma usually reveals itself as episodes---'  It would be better if you had:  Asthma usually presents as episodes or attacks. Asthma  symptoms include wheezing---- which are often triggered by respiratory tract infections----"   

Depressive disorders or depression is a medical condition are not really a symptom (line 59, page 2). additionally what is a borderline state of depression, no definition available.

The results needs to be listed with the important findings first,

70% of the COPD patients were retired, (line 118) based on this statement, are the text on lines 156 and 157  a novel finding? additionally what is teh difference between pensioners and retired from employment, it is not clear. Need an explanation. 

In the conclusion, there is a lot of redundancy, please  go through it and just include only what your study conclusion is.

Author Response

REVIEWER 1

The results presented a needs to be clearer, please see the highlighted text and comments it in the document. 

According to the comments embedded in the manuscript:

Line 66: changed to ‘depression’

Line 107: professional statistician who conducted the analysis declared that it was prepared using R program version 3.6.2

Line 150: We are sorry, but the ’what %’ suggestion is quite unclear for us. Nonetheless, for more clarity we replaced ‘groups of’ with ‘in general’ in the beginning of the sentence.

Line 156: Since 2017 retirement age has been approved to be 60 for women and 65 for men. According to the national reports mean age of retirement in 2019 was 62,1 overall (64,6 for men and 60,7 for women). In 2021 the statistics were similar and the mean age was 62,4. Hence there is rather significant proportion of retired and retired employees in our study.

Line 224: the authors of the cited study do not clarify the meaning of ‘outpatients’, but we assume this means patients treated in the clinics (not-hospitalized)

Please make the definitions and the symptoms clear, also make sure the words used are appropriate in the text For example, page 1 line 44: 'Asthma usually reveals itself as episodes---'  It would be better if you had:  Asthma usually presents as episodes or attacks. Asthma  symptoms include wheezing---- which are often triggered by respiratory tract infections----" 

Thank you, the sentence from line 44 has been changed.  

Depressive disorders or depression is a medical condition are not really a symptom (line 59, page 2). additionally what is a borderline state of depression, no definition available.

In line 59 instead of changing ‘depressive disorders’ to ‘depression’ word ‘symptoms’ was changed to ‘problems’, therefore according to your suggestion depressive disorders are no longer considered as symptoms in the manuscript.

The phrase ‘borderline state’ has been removed and replaced with ‘borderline abnormal’ to make the term more accurate. Also highest scores are now defined as ‘abnormal’. Thank you for your suggestion.

The results needs to be listed with the important findings first,

Thank you for a valid comment. Taking into account the vast data we attempted to characterize the study group in the beginning. The remaining paragraphs have been changed to provide the important findings first, while in the next presenting less applicable results.

70% of the COPD patients were retired, (line 118) based on this statement, are the text on lines 156 and 157  a novel finding? additionally what is teh difference between pensioners and retired from employment, it is not clear. Need an explanation. 

The number in line 118 is merely informative in terms of group characteristics. Lines 156 and 157 provide information about anxiety in regards to the employment status.

A mistranslation ‘retired from employment’ has been changed into ‘retired employees’.

In the conclusion, there is a lot of redundancy, please  go through it and just include only what your study conclusion is.

The conclusion has been shortened nearly by a half.

Dear Reviewer,

Thank you very much for your review. It allowed us to improve the quality of our manuscript.

Yours sincerely

Mariusz Chabowski

Reviewer 2 Report

Introduction:

-       Line 52: It is now considered asthma-COPD overlap, the word syndrome was removed from the updated guidelines. Please correct and also the associated abbreviation.

-       I suggest the author to modify their introduction. For the 1st and 2nd paragraph, they could be summarized in one paragraph. Also, I suggest the authors to add a paragraph on the extrapulmonary manifestation of both diseases especially COPD as still fatigue and psychological changes are parts of extra-pulmonary manifestations. 

Methods

-       Please clearly describe the study design. There is nothing mentioned about this. Is this retrospective or prospective study? Please clarify 

-       What are the inclusion and exclusion criteria of the participants?

Results 

-       The authors presented the data with mean / SD, mean and quartiles in tables. The data should be tested to normal distribution and according the data presented, not both presentation for all data. Please correct with description in the methods (statistical analysis section).

-       What is the severity of the diseases? Did the patients have spirometry? This data should be presented in the results.

-       Also, for the asthma, what is the degree of disease control? 

-       In the marital status: what did mean in relationship? This is vague incorrect term, please specify as single, married, divorced and widow.

-       Please add the level of education in table 1.

-       In the text (1st paragraph), the authors should add the p values and describe clearly the significant difference.

-       In table 1: what do the authors mean by Summary (the 4th column)? Also, what comparison do p valuespecify? Please add a note below the table for verification.

-       In the smoking history (table 1): there is no ex-smoker, could you please explain why?

-       What is the difference between table 3 and 4? What is the benefit to describe in numerical scales and categories?

-       I cannot find the importance of the correlation between MEIS and HADS. Based on the methods provided by the authors, I understood that these two scales for anxiety and depression. Why did the authors correlate both scales (table 5 and figure 1)? I was expected to correlate these scales between the disease severity and disease duration.

-       Lines 154-165: this data should be presented as table with clear description and showing the p value. Please specify and provide figures.

Discussion

-       The 1st paragraph is too long. I cannot understand this is just for summarization of results or provide explanation. I suggest the authors to mention only the important data in first paragraph in just few lines before go for explanation.

-       The discussion should be modified according to results modification.

General

-       What is the novelty of the current study? The items of depression and psychological disturbance have been studied previously in many studies. 

Author Response

REVIEWER 2

Introduction:

-       Line 52: It is now considered asthma-COPD overlap, the word syndrome was removed from the updated guidelines. Please correct and also the associated abbreviation.

Thank you, we applied the changes according to your suggestion.

-       I suggest the author to modify their introduction. For the 1st and 2nd paragraph, they could be summarized in one paragraph. Also, I suggest the authors to add a paragraph on the extrapulmonary manifestation of both diseases especially COPD as still fatigue and psychological changes are parts of extra-pulmonary manifestations. 

The paragraphs were joined into one paragraph. Additional information are now included in the introduction.

Methods

-       Please clearly describe the study design. There is nothing mentioned about this. Is this retrospective or prospective study? Please clarify 

It is a prospective questionnaire-based study with sociodemographic data acquired from patients’ medical records. The specific information has been added in the beginning of the ‘Methods’ section.

-       What are the inclusion and exclusion criteria of the participants?

The criteria are now included in the Methods section – also provided below.

‘The inclusion criteria were: informed written consent to participate in the study, diagnosis of COPD or asthma or ACO, age above 18 years. Patients below 18 years, with concomitance of severe chronic diseases that could interfere with the study (neoplastic diseases of the lungs, respiratory failure, exacerbation of chronic disease), unable to complete the questionnaire independently were excluded from the study.’

Results 

-       The authors presented the data with mean / SD, mean and quartiles in tables. The data should be tested to normal distribution and according the data presented, not both presentation for all data. Please correct with description in the methods (statistical analysis section).

Each variable was tested for normal distribution, but none proved to have normal distribution. However, according to our statistician’s suggestion, we would prefer to keep all values, because on the other hand deleting any of those brings about questions about missing data. Thank you for your comment.

-       What is the severity of the diseases? Did the patients have spirometry? This data should be presented in the results. Also, for the asthma, what is the degree of disease control? 

Unfortunately, these data were not included in the study. It is a limitation of this study. Thank you for pointing this out. It is now in limitations.

The number of the hospital admissions due to complications is the only information about disease control in our study.

-       In the marital status: what did mean in relationship? This is vague incorrect term, please specify as single, married, divorced and widow.

Thank you for pointing this out. However, the data does not specify the marital status beyond in/not in a relationship. Our aim at the point of the study design was to test the hypothesis about influence of being in formal or informal relationship vs not being in a relationship on studied parameters as this can greatly impact the level of patients’ social support.

-       Please add the level of education in table 1.

The education section has been added to the Table 1.

-       In the text (1st paragraph), the authors should add the p values and describe clearly the significant difference.

The significant difference was denoted by p-value in the last paragraph (statistical analysis) of the ‘Methods’ section. P-values regarding data presented in the first paragraph of the ‘Results’ section are available in Table 1. Adding p-values to the text of the manuscript would be – in our opinion - an unnecessary repetition.

-       In table 1: what do the authors mean by Summary (the 4th column)? Also, what comparison do p value specify? Please add a note below the table for verification.

‘Summary’ was changed to ‘Total’ and p-values are now described below the table. We hope this brings clarity to the numbers.

-       In the smoking history (table 1): there is no ex-smoker, could you please explain why?

Patients were not asked that question, so this information is not available. Thank you for your comment. We will surely keep this in mind when planning our future studies.

-       What is the difference between table 3 and 4? What is the benefit to describe in numerical scales and categories?

Table 3. presents mean, SD, median and quartiles of patients’ scores in points, which does not inform what was the distribution of specific scores’ range among studied groups. The latter is provided in Table 4. We believe that these two tables complement each other.

-       I cannot find the importance of the correlation between MEIS and HADS. Based on the methods provided by the authors, I understood that these two scales for anxiety and depression. Why did the authors correlate both scales (table 5 and figure 1)? I was expected to correlate these scales between the disease severity and disease duration.

The correlations are important as the scores of each subscale or the total scores of either MFIS or HADS correlate significantly with any subscale or total score of  the compared scales. This brings a conclusion that the increase of any subscale’s score (meaning the increase of intensity of a problem) entails an increase in others.

-       Lines 154-165: this data should be presented as table with clear description and showing the p value. Please specify and provide figures.

These data were not provided as a table since gathering all the data - to which these lines relate – in a table would make it enormously large, complicated and indecipherable. We concluded only those information that proved significant correlation. For the sake of simplicity we prefer not to create another table. However, the r- and p-values are now added to the manuscript.

Additionally a mistranslation ‘retired from employment’ has been changed into ‘retired employees’.

Discussion

-       The 1st paragraph is too long. I cannot understand this is just for summarization of results or provide explanation. I suggest the authors to mention only the important data in first paragraph in just few lines before go for explanation.
The discussion should be modified according to results modification.

 The discussion has been shortened. Thank you for this advice.

General

-       What is the novelty of the current study? The items of depression and psychological disturbance have been studied previously in many studies. 

First and foremost this study confirms a significant positive relation of HADS and MFIS scores, thereby providing evidence that ailments described by both scales influence each other. The preceding finding has an important clinical implication that in order to improve any aspect of patient’s well-being (regarding the studied scales) an approach aiming at all the problems should be administered.

Dear Reviewer,

Thank you for your comments. We are confident that your suggestions will greatly contribute to improving the quality of our research. We hope our answers are thorough and the changes applied according to your comments bring more clarity to our manuscript.
